# Transformational Leadership and Emotional Labor: The Mediation Effects of Psychological Empowerment

**DOI:** 10.3390/ijerph20021030

**Published:** 2023-01-06

**Authors:** Pengfei Cheng, Zhuangzi Liu, Linfei Zhou

**Affiliations:** School of Economics and Management, Xi’an University of Technology, Xi’an 710054, China

**Keywords:** emotional labor, transformational leadership, psychological empowerment

## Abstract

In order to survive the fiercer competition, more and more service firms emphasize front-line employees’ role of creating excellent customer experience by displaying positive emotions during the service interactions. However, the underlying mechanisms for the relationship between transformational leadership and front-line employees’ emotional labor remain unclear. Drawing upon the conservation of resources (COR) theory, this study develops a conceptual model in which transformational leadership influences front-line employees’ emotional labor through the mediator of psychological empowerment. By collecting data from 436 employees in five call centers, we tested our model and hypotheses through PROCESS 3.3 macro for SPSS developed by Hayes. The results show that transformational leadership shows positive and negative effects on deep acting and surface acting, respectively. The positive effect on deep acting is partially mediated by psychological empowerment, while the negative effect on surface acting is fully mediated by psychological empowerment. Specifically, two dimensions of psychological empowerment (impact, self-efficacy) play negative mediating roles between transformational leadership and surface acting, while impact, self-determination, and self-efficacy play positive mediating roles of transformational leadership and deep acting. The findings advance our understanding about how transformational leadership influences front-line employees’ emotional labor by introducing psychological empowerment as a mediator.

## 1. Introduction

As the fast rise of the service economy in China and the competition becomes fiercer, more and more service firms are coming to realize the pivotal role of front-line employees in creating an excellent service experience. Front-line employees are required to display appropriate emotions during their interactions with customers [1]. In her seminal work, Hochschild [2] defined emotional labor as front-line employees’ expression of expected emotions during service encounters. Employees can take two different strategies to perform emotional labor: surface acting and deep acting. Surface acting involves simulating emotions that are not actually felt. In contrast, deep acting involves attempts to actually experience the emotions one is required to display. Considerable evidence indicates that deep acting is more likely to lead to positive outcomes than surface acting [3,4,5,6].

Given the interactive nature of service process, how front-line employees display their emotions during service encounters determines the service performance. A vast body of research has given interest to identifying the antecedents of the emotional labor. One of the research streams on the antecedents of emotional labor focuses on customer behaviors, such as customer incivility [7] and customer participation [8]. However, another stream of research focuses on front-line employees’ individual differences, such as dispositional traits [9], self-verification striving [10], and positive affectivity [11]. Due to the tight connection between leaders and front-line employees, a growing number of studies recently have shown interest in exploring the effects of leadership on emotional labor [12,13]. Transformational leadership is defined as the ability of managers to provide followers with challenging goals, motivating them to perform beyond the specified expectations [14]. Prior research suggested that transformational leadership has important effects on emotional labor [13,15]. However, the mechanism between transformational leadership and emotional labor has received little attention in previous literature.

To understand how different emotional labor strategies of front-line employees are influenced by transformational leaders, we examined the mediation effect psychological empowerment on the relationship between transformational leadership and emotional labor. Psychological empowerment, which involves employees’ active orientation to their work role, is an important determinant of employee behaviors [16]. Through being empowered, front-line employees could get more internal resources in their job [17]. Previous research has shown that psychological empowerment could mediate the effects of the transformational leadership on employees’ work outcomes [18,19,20,21]. However, to our best knowledge, few empirical studies have examined the potential mediating mechanisms of psychological empowerment through which transformational leadership influences front-line employees’ emotional labor. Therefore, to address this gap, the purpose of this study is to unveil the underlying processes responsible for the effects of transformational leadership on front-line employees’ emotional labor by introducing psychological empowerment as mediator.

To summarize, we proposed a mediation model in which transformational leadership influences front-line employees’ emotional labor by improving psychological empowerment. Different from previous studies which mainly consider psychological empowerment as a unidimensional construct [22,23,24], the current study tries to further explore how transformational leadership influences emotional labor through four dimensions of psychological empowerment differently: meaning, impact, self-determination, and self-efficacy. By doing so, we may further explain the linkage between transformational leadership and employee’s emotional labor and shed light on how managers might effectively interfere with front-line employees’ emotional labor. In the following sections, we explicate the theorical background of transformational leadership and emotional labor and consider building bridges from four dimensions of psychological empowerment to the relationship between the prior two. We then test our hypotheses using a sample of 436 employees. Finally, we report the results and discuss the implications for theory and practice, offering constructive and specific inspiration for the service industry.

## 2. Theoretical Model and Hypotheses

### 2.1. Emotional Labor of Front-Line Employees

Emotional labor refers to front-line employees obeying their organizational expectation by regulating and displaying appropriate emotions during service encounters [25]. According to Hochschild [2], employees perform emotional labor by taking two different strategies: surface acting and deep acting. In surface acting, front-line employees comply with display rules by suppressing their inner feelings and expressing feigned rather than genuine emotions. Conversely, in deep acting, front-line employees try to actually experience the emotions they are required to display [26]. According to Ashforth and Humphrey [27], employees devote greater psychic efforts to perform deep acting than to perform surface acting. Deep acting requires front-line employees to actively invoke their thoughts, images, and memories to alter how they feel. Many previous studies have indicated that these two different emotional labor strategies lead to different employee well-being and service performance [28,29,30]. Generally speaking, surface acting results in negative employee and organization outcomes, including burnout [31], low job satisfaction [32], and more stress [33]. In contrast, the consequences of deep acting are more positive. Deep acting can reduce employee emotional exhaustion and enhance emotional performance [34,35]. Hence, just as Pugh et al. [36] calls for enhancing the practice of ‘good’ emotional labor in employees by promoting the use of deep acting and discouraging the use of surface acting, researchers are increasingly shifting their attention to explorer the antecedents of emotional labor. In order to help service firms effectively manage their front-line employees’ emotional labor, many researchers have identified antecedents of emotional labor from an organizational perspective, such as organizational fairness [37] and leadership style [15]. Noticeably missing from research attention is the mechanism by which transformational leadership influences emotional labor, despite Chi et al. [38] having indicated that transformational leadership is positively related to service employees’ deep acting.

### 2.2. Transformational Leadership

Transformational leadership refers to a set of behavior which elevate the capacity of followers through transforming their values, beliefs, and attitudes [39]. Previous studies have identified four dimensions of transformational leadership: charisma, inspirational motivation, intellectual stimulation, and individualized consideration [14]. Specifically, charisma refers to the ability of leaders to generate trust, admiration, and emulation of followers [40]. Inspirational motivation is defined as an encouragement behavior of leaders to motivate their followers to understand the vision of organization and exceed the established performance standards [14,40]. Intellectual stimulation is a behavior that leaders provoke followers to challenge existing assumptions and to face the problems with more innovative and creative in their work [18,41]. Individualized consideration involves treating each follower as a unique one, providing them with learning opportunities, and offering personal attention on their needs and concerns [42]. Previous studies have indicated that transformational leadership might lead to positive employee and organizational outcomes, such as high job satisfaction [43], organizational citizenship behavior [44], and organizational culture [45,46,47]. From the perspective of employees, Saira [18] suggested that transformational leaders can improve employees’ organizational citizenship behavior and reduce their turnover intention. Her research further illustrates that transformational leadership’s effects on employee behavior (OCB) are mediated by psychological empowerment. Similar findings was provided by Stanescu [21] that empowerment acts is necessary and effective processes for transformational leaders to foster innovation behaviors among followers. These findings imply that psychological empowerment plays an important mediating role in transformational leadership’s effects on employees’ behavior. Thus, it is reasonable to infer that transformational leadership influences front-line employees’ emotional labor through psychological empowerment.

### 2.3. Transformational Leadership’s Effects on Psychological Empowerment

Transformational leaders encourage subordinate self- development, provide a vision for the future, and pay attention to the subordinate’s needs by exhibiting four kinds of behaviors: idealized influence, intellectual stimulation, inspirational motivation, and individualized consideration [19,48,49]. This people-orientated leadership style will foster the development of employees’ positive psychological and organizational behavior [50]. Since transformational leaders rely more on innovative manners to deal with problems of their subordinates, the positive attitude of employees toward leaders and organizations are fostered, such as trust [51]. Thus, transformational leadership contributes front-line employees to generating intrinsic motivation and resources in some form, such as psychological empowerment.

According to the transformational leadership theory, empowerment has been recognized as a positive strategic management practice. Transformational leaders make it possible for employees to participate in the process of decision-making, cooperation, and idea generation, which could make them feel more empowered in their work [21]. Thus, subordinates of transformational leaders believe in the positive impact they could make to the organization [18]. In addition, research has pointed that psychological empowerment is vulnerable to leadership [20,24,52,53]. The research of Aydogmus [16] shows that employees will feel more psychologically empowered when they perceive their leaders as transformational. In other words, these behaviors from transformational leadership will lead to a certain external stimulus, and this kind of environmental stimulation will form a perception of psychological empowerment through the internal evaluation of employees [54]. Therefore, it is inferred that transformational leadership—as an external stimulus that pays attention to employee development—will affect psychological empowerment of employees.

Psychological empowerment refers to the extent of individual’s perception of empowerment [55]. Previous studies have shown that employees’ perceptions of psychological empowerment consist of four dimensions: meaning, impact, self-determination, and self-efficacy [56]. Specifically, meaning suggests that employees believe in the significance of their work and the alignment between their work goals and their personal goals [55]. Impact is a degree that individuals influence organizational outcomes, such as strategies, operations, and management [55]. Self-determination reflects a control of decision-making, emotions and behaviors, or the extent of freedom employees have in their work [21,57]. Self-efficacy refers to one’s confidence of his/her capacity to perform a job with competence [55,56]. First, inspirational motivation of transformational leaders motivates employees to transcend the current standard of organization, which makes them view more value and meaning in their job [51]. Meanwhile, individualized consideration by transformational leaders will heighten personal development of employees [58]. According to the conservation of resources (COR) theory, these conditions increase job resource of employees, which are related to the personal growth of them [59], leading more positive attitudes towards organization, and then trigger the sense of meaning for them. Second, in the preceding passages, transformational leaders actively encourage employees to believe their ability [18]. On the one hand, these attempts create true feelings for employees about their contributions to an organization [60]. This feeling promotes them to have beliefs in their capability to have a great influence on the organization. On the other hand, encouragement from transformational leaders increases the confidence levels of employees, resulting in their self-efficacy [61]. Finally, intellectual stimulation of transformational leaders encourages their employees to challenge the status quo, to be innovative, and to take more responsibility in their work [53]. Consequently, employees will feel more empowered to deal with problems with a high level of freedom [20,51], enhancing their self-determination. Thus, we propose:

**H1a.** 
*Transformational leadership is positively related to meaning.*


**H1b.** 
*Transformational leadership is positively related to impact.*


**H1c.** 
*Transformational leadership is positively related to self-determination.*


**H1d.** 
*Transformational leadership is positively related to self-efficacy.*


### 2.4. Psychological Empowerment’s Effects on Emotional Labor

#### 2.4.1. Meaning’s Effects on Emotional Labor

According Grandey et al. [62], front-line employees’ emotional labor, as a kind of discretionary behavior, is mainly derived by employees’ intrinsic motivations. The meaning of the job, which serves as an important promotor of one’s intrinsic motivation, can motivate front-line employees to fulfill their job duties [63]. For example, when front-line employees can derive meaning from their daily work, they are more likely to internalize their organizational emotion display rules. Hence, we can infer that front-line employees who can get meaning from their jobs are prone to take deep acting rather than surface acting to comply with display rules when interacting with customers. Furthermore, the meaning of a job can bring front-line employees positive emotions [64]. With a more positive emotional state, it will take front-line employees much less effort to adjust their real feelings. According to the conservation of resources (COR) theory, positive emotions, as a kind of personal resource, can guarantee front-line employees to take deep acting, which creates more personal resources [65]. In addition, front-line employees with a high level of intrinsic motivation have a high level of job competence, such as cognitive flexibility and emotional regulation abilities [66]. These kinds of abilities makes front-line employees easer to induce themselves to feel the expected emotions, namely performing deep acting.

Conversely, organizational emotional display rules could be a burden for front-line employees who cannot get meaning from their job, bringing them more cognitive limitations. For these employees, they must always pay attention to their internal and external contradictory emotional states and spend energy restraining negative emotions. On the other hand, weakness of intrinsic motivation may also make front-line employees unaware of how to actively adjust their cognitions when facing unreasonable treatment from customers. Under this condition, front-line employees are prone to take surface acting to deal with the pressure from customers. Thus, we propose:

**H2a.** 
*Meaning is positively related to deep acting.*


**H2b.** *Meaning is negatively related to surface acting*.

#### 2.4.2. Impact’s Effects on Emotional Labor

In the workplace, how employees perceive the desired effects of their behaviors and the extent to which it affects their organization is defined as work impact [67]. Employees who are aware of the impact of their work have stronger feelings of responsibility towards the organization. They pay more attention to their emotions expressed during the process of interaction with customers. Such responsibility makes front-line employees actively internalize organizational service norms and values. Especially in tough times, this can help front-line employees overcome negative emotions and perform deep acting. In contrast, front-line employees who perceive low level of their work impact usually have a weak feeling of responsibility towards their organization. As a result, during their daily work, they are not prone to genuinely alter their inner feelings and pretend to feel desired emotions, namely surface acting. Thus, we propose:

**H3a.** 
*Impact is positively related to deep acting.*


**H3b.** *Impact is negatively related to surface acting*.

#### 2.4.3. Self-Determination’s Effects on Emotional Labor

Self-determination means a selective cognition of individuals in initiating and regulating their own behavior [68], reflecting front-line employees’ level of autonomy in the workplace [69]. Previous researches have indicated that employees with high self-determination are of higher levels of concentration, initiative, and resilience [55]. Under this condition, front-line employees will devote much more effort to modify their inner feelings so as to display genuine, organization-desired emotions. Additionally, for the front-line employees with high autonomy, they will feedback their organizations with high identification, attachment, and loyalty [70], which will lead front-line employees to take more positive actions, such as deep acting to cope with job problems [71].

In contrast, it is difficult for employees with low self-determination to get a high sense of control over their job. Under the condition of low sense of control, front-line employees perceive high uncertainty and take passive manners to cope with job roles, such as surface acting [6]. Furthermore, a low level of self-determination involves feeling helpless and a lack of psychological resources. According to the conservation of resources (COR) theory, front-line employees with low level personal resources are more likely to take surface acting, which requires less personal resources to cope with their job roles [19]. Thus, we propose:

**H4a.** 
*Self-determination is positively related to deep acting.*


**H4b.** *Self-determination is negatively related to surface acting*.

#### 2.4.4. Self-Efficacy’s Effects on Emotional Labor

Self-efficacy refers to the extent of belief or confidence that one can perform work activities successfully [72,73]. As a form of psychological resources, self-efficacy causes positive psychological states, including felt responsibility, which motivates front-line employees to invest more efforts to perform their job roles [74]. According to the conservation of resources (COR) theory, front-line employees working in a resourceful environment are more likely to take deep acting, which duplets more resources than surface acting. Additionally, previous research has indicated that employees with high self-efficacy feel more confident to cope with customer demanding and make a positive appraisal of their work environment [70]. In other words, self-efficacy causes front-line employees positive psychological states, such as experiencing positive emotions. Under this condition, it is much easier for front-line employees to regulate their inner feelings (deep acting) to comply with the display rules than the condition of experiencing negative emotions [75].

Conversely, for front-line employees with low self-efficacy, they will underestimate their capabilities, which makes them experience negative emotions, such as stress and anxiety [76]. Negative emotions, which violate the basic service norm, make front-line employees have to suppress their inner feelings and pretend to experience positive emotions, namely surface acting. Furthermore, self-efficacy, as a foundation of intrinsic motivation, drives front-line employees to work hard and expect desired performance. However, for front-line employees with low self-efficacy, they are prone to take less effortful faking process without altering how they feel, namely surface acting. See Figure 1 for a model overview. Hence, we propose:

**H5a.** 
*Self-efficacy is positively related to deep acting.*


**H5b.** *Self-efficacy is negatively related to surface acting*.

## 3. Methodology

### 3.1. Sampling and Data Collection

In order to test the hypotheses, we carried out a survey to collect data from front-line employees of call centers. The call center was selected as our study context for its character of intense employee–customer interaction, which requires front-line employees to display positive emotions during the service encounter [77]. After selecting five call centers located in northwestern China, a total of 500 questionnaires were distributed on site to front-line employees with 100 questionnaires for each call center. The questionnaire was totally anonymous and took participants about 15 min to complete. When participants finished the survey, they were required to seal the questionnaire in an envelope by themselves and give it to the research assistants on site. Finally, 483 were returned. After deleting 47 incomplete ones, we obtained 436 valid questionnaires, indicating a response rate of 87.2%. The sample consisted of 436 employees (man = 30, woman = 406). The number of employees between 25 and 30 years was the largest (23.6%). Overall, 78% of employees had received a junior college education or above. In terms of tenure, the largest number of groups is one to three years (36.7%).

### 3.2. Measure Operationalization

For constructs involved in this study, we use previously established scales to measure. Because the original scales were developed in English, following the translation and back-translation procedures recommended by Brislin [77], we translated these measurement scales into Chinese. All measures except for demographic variables were reported on a five-point Likert scale, ranging from 1 (completely disagree) to 5 (completely agree).

*Transformational Leadership*. Transformational leadership was measured by a 20-item scale—the Multifactor Leadership Questionnaire (MLQ)—adapted from Bass and Avolio’s [48]. MLQ consists of four correlated dimensions: charisma (8-item), inspirational motivation (4-item), intellectual stimulation (4-item), and individualized consideration (4-item).

*Psychological Empowerment*. Adapting from Spreitzer [56], psychological empowerment was measured via 12 items tapping four dimensions: meaning, impact, self-determination, and self-efficacy. Each dimension includes three items.

*Emotional labor.* Front-line employees’ emotional labor was measured by using two 3-item scales, adapted from Brotheridge and Lee [78]. Both deep acting and surface acting were measured by three items.

*Control Variables.* To exclude potential confounding effects of front-employee demographics on emotional labor, we controlled for gender (0 = female; 1 = male), age (1 = under 21; 2 = 21 to 25; 3 = 25 to 30; 4 = 30 to 35; 5 = above 35), education (1 = Junior high school and below; 2 = High school; 3 = Undergraduate; 4 = Postgraduate and above), and tenure (1 = less than 1 year; 2 = 1 to 3 years; 3 = 3 to 5 years; 4 = 5 years and above).

### 3.3. Descriptive Statistics and Intercorrelations

The means, standard deviations, reliabilities, AVE, and correlations among the research variables are presented in Table 1. The Alpha coefficients for all constructs ranging from 0.862 to 0.932 indicate that the reliability is acceptable. Additionally, it can be seen that the significant correlations in the matrix are between the variables most proximal to each other in the hypothesized model.

### 3.4. Confirmatory Factor Analysis and Common Method Bias Testing

To examine the discriminant validity of the seven latent constructs, we first conducted a series of confirmatory factor analysis (CFA) with Amos 26.0 by utilizing maximum-likelihood estimation to test the measurement model (see Table 2). A total of four measurement models were compared by several fit indices: seven factors, three factors, two factors, and one factor models. The results suggest that the seven factors model fits the data best ($\chi^{2}$/ df = 2.695, root mean square of approximation [RMSEA] = 0.062, comparative fit index [CFI] = 0.904, Tucker–Lewis index [TFI] = 0.894). Thus, the discriminant validity of the seven latent constructs is acceptable. Furthermore, the average variance extracted (AVE) of all constructs were between 0.642 and 0.879 (see Table 1), which exceeded the cut-off value of 0.5, indicating that the convergent validity of scales was acceptable.

Since we used self-report methods to collect data, there may be common method bias, which may result in spurious relationships among the variables [79]. The results of Harman’s single factor test have demonstrated that the first factor only accounts for 32.070% of the total variance, which indicated that common method bias was not present.

## 4. Results

To test the mediation effects of psychological empowerment on the relationships between transformational leadership and emotional labor, this study ran a series of regressions using the SPSS PROCESS macro developed by Hayes [80]. Specifically, following the suggestion of Hayes (2018), this study performed bootstrapping analysis with 5000 replications, which can generate 95% confidence intervals (CI) for total effects, direct effects, and indirect effects. The significance of mediating effect depends on whether CI contains zero or not.

First, this study tested the effects of transformational leadership (TL) on deep acting (DA) via the four dimensions of psychological empowerment. The results of mediation tests are summarized in Table 3. Coefficients of M-1 indicate that transformational leadership is positively related to deep acting (β = 0.465, *p* < 0.001). As mentioned earlier, we have hypothesized that transformational leadership influences deep acting through the meaning (Mean), impact (IMP), self-determination (SD), and self-efficacy (SE). Therefore, we examined the effects of transformational leadership on meaning, impact, self-determination, and self-efficacy, in M-2 to M-5. As expected, transformational leadership was found to be positively associated with meaning (β = 0.431, *p* < 0.001), impact (β = 0.406, *p* < 0.001), self-determination (β = 0.462, *p* < 0.001), and self-efficacy (β = 0.417, *p* < 0.001), supporting H1a to H1d. These findings demonstrate that transformational leaders in service firms could boost front-line employees’ experience of psychological empowerment, which is consistent with the findings of prior studies.

Then, the results of M-6, which regressed deep acting on transformational leadership and four mediators (meaning, impact, self-determination, and self-efficacy) simultaneously, show that transformational leadership’s positive effect (β = 0.304, *p* < 0.001) on deep acting remains. Three dimensions of psychological empowerment (impact (β = 0.106, *p* < 0.05), self-determination (β = 0.191, *p* < 0.001), and self-efficacy (β = 0.111, *p* < 0.05) are positively related to deep acting, supporting H3a, H4a, and H5a. However, meaning’s effect on deep acting is not significant, rejecting H2a. The results of bootstrap analyses, summarized in Table 4, suggest that the CIs for impact (CI = 0.005, 0.116), self-determination (CI = 0.040, 0.198), and self-efficacy (CI = 0.006, 0.123) do not include zero and CI for meaning (CI = −0.075, 0.038) include zero. Taken together, the effect of transformational leadership on deep acting is partially mediated by impact, self-determination, and self-efficacy.

Secondly, we tested the effects of transformational leadership (TL) on surface acting (SA) via the four dimensions of psychological empowerment. As presented in Table 3, coefficients of M-7 indicate that transformational leadership shows a negative effect on surface acting (β = −0.250, *p* < 0.001). Then, coefficients of M-8 suggested that after entering four mediators (meaning, impact, self-determination, and self-efficacy), the effect of transformational leadership on surface acting becomes non-significant (β = −0.094, *p* > 0.05). In addition, impact (β = −0.172, *p* < 0.01) and self-efficacy (β = −0.232, *p* < 0.001) are negatively related to surface acting, supporting H3b and H5b. However, neither meaning (β = −0.059, *p* > 0.05) or self-determination’s effects (β = 0.079, *p* > 0.05) on surface acting are significant, rejecting H2b and H4b. In Table 5, the results of bootstrap analyses indicate that CIs for impact (CI =−0.158, −0.21) and self-efficacy (CI = −0.190, −0.055) do not include zero, while CIs for meaning (CI = −0.089, 0.030) and self-determination (CI = −0.029, 0.124) include zero. In addition, the direct effects of transformational leadership were not significant (CI = −0.246, 0.021). Taken together, the effect of transformational leadership on surface acting is fully mediated by impact and self-efficacy.

To summarize, for the effects of transformational leadership on emotional labor, all of our hypotheses are supported except H2a, H2b, and H4b (see Table 6). The results indicate that transformational leadership influences deep acting and surface acting through distinct mediating paths. Specifically, transformational leadership’s positive effects on front-line employees’ deep acting are partially mediated by impact, self-determination, and self-efficacy. In contrast, transformational leadership’s negative effects on front-line employees’ surface acting are fully mediated by impact and self-efficacy.

## 5. Discussion

The current study proposed and tested a conceptual model to explore the mechanism by which transformational leadership influences the emotional labor strategies of front-line employees. Our results demonstrate that transformational leadership exerts positive and negative effects on deep acting and surface acting, respectively. Psychological empowerment serves as a mediator of the linkage between transformational leadership and two emotional labor strategies. Specifically, the positive effect of transformational leadership on deep acting is partially mediated by three dimensions of psychological empowerment: impact, self-determination, and self-efficacy. The negative effect of transformational leadership on surface acting is fully mediated by two dimensions of psychological empowerment: impact and self-efficacy.

### 5.1. Contributions to Theory

Firstly, the results of this study help us better understand the psychological process by which organizational factors (e.g., transformational leadership) drive front-line employees’ emotional labor. Although previous studies have paid attention to the effects of transformational leadership on front-line employees’ emotional labor, such as Luo et al. [13], the psychological mechanism of these effects remain unclear. Grandey et al. [62] mentioned in their work that front-line employees’ emotional labor is a form of discretionary behavior, which is driven more by intrinsic motivation. It is therefore reasonable to explain how transformational leadership can effectively influence front-line employees’ emotional labor from an intrinsic motivation perspective. The findings of this study uncover the “black box” between transformational leadership and emotional labor by introducing psychological empowerment as a mediator. This study supplements Luo and Guchait’s work [13], which merely examines the direct effect of transformational leadership on emotional labor.

Secondly, the results of this study contribute to the literature on transformational leadership as well. Despite an array of prior studies that have demonstrated that transformational leadership is an effective leadership style for bootstring proactive employee behaviors, such as OCB [81] and innovation behaviors [58,60], only a few studies have recently turned their interests to examine transformational leadership’s effects on front-line employees’ emotional labor [13], which is a very common form of proactive behavior during their daily work. Not only does this study confirm transformational leadership’s effects on front-line employees’ emotional labor in the context of service, but this study also illustrates the psychological process of these effects by examining psychological empowerment’s mediating effects. These findings answer Siangchokyoo’s [82] call for more detailed studies to examine the role of empowerment plays during the link of transformational leadership and employees’ behavior.

Thirdly, this study extends research on psychological empowerment. Although prior studies have confirmed that psychological empowerment, as an important psychological construct, is a typical consequence of transformational leadership [82], only a few studies have further extended this link to employees’ behavior. The results of this study confirm the existence of the logical path: transformational leadership–psychological empowerment–emotional labor. Furthermore, by considering psychological empowerment as a four-dimension construct (meaning, impact, self-determination, and self-efficacy), this study distinguishes the different roles of four dimensions of psychological empowerment play during the link of transformational leadership and different emotional labor strategies. It therefore pushes forward our understanding of the distinctiveness of four psychological empowerment dimensions.

### 5.2. Managerial Implications

As competition becomes fiercer [83], service managers are increasingly highlighting the emotions front-line employees display during their interaction with customers. However, given the discretionary nature of front-line employees’ emotional labor, it is a big challenge for managers to effectively influence their subordinates’ emotional labor. The results of this study may present interesting insights for managers in intervening in their subordinates’ emotional labor.

Firstly, service firms have to realize the importance of a leader in influencing front-line employees’ emotional labor. According to the findings of this study, transformational leadership is an ideal leadership style for front-line employees. Therefore, during the leader recruiting or promotion process, it is necessary to set criteria considering candidates’ competence or personality, which make them easier to perform transformational leadership behaviors. By doing so, service firms can guarantee that leaders are suitable for their job and can effectively exert influence on front-line employees.

Secondly, service managers should alter their leadership style to transformational leadership, which has been suggested to promote front-line employees’ deep acting and reduce surface acting. Therefore, training programs for service managers should focus on the skills of engaging transformational leadership behaviors. According to the definition of transformational leadership, typical transformational leadership behaviors include idealized influence, intellectual stimulation, inspirational motivation, and individualized consideration [41]. Once leaders take these behaviors in their routine work, the front-line employees tend to take deep acting rather than surface acting during service encounters. Furthermore, organizational culture also can foster a climate suitable for transformational leadership. If the organizational culture encourages employee self-growth, providing employee future vision, and caring for employee needs and well-being, the leaders are prone to take transformational leadership to comply with the organizational culture.

Thirdly, efforts should be devoted to enhancing front-line employees’ psychological empowerment. The results of this study show that psychological empowerment plays an important role in the link of transformational leadership and emotional labor. In order to enhance front-line employees’ psychological empowerment, service firms should especially improve employees’ sense of impact. For example, service firms can send signals of caring and valuating employees’ contribution by providing positive feedback, such as rewarding. To improve front-line employees’ self-determination, managers can invite front-line employees to take part in the decision-making. Additionally, enhancing front-line employees’ job autonomy through empowerment is an effective way to improve front-line employees’ self-determination. In addition, enhancing self-efficacy is necessary to eliminate front-line employees’ surface acting. Managers should encourage employees to set challenging goals and provide support to help them to overcome difficulties and complete the tasks.

### 5.3. Limitations and Future Research

Although this study makes theoretical and practical contributions, there are also several limitations. First, the research context of this study focusses on call centers. Even though emotional labor is very common for call center employees, employees’ emotional labor strategies or efforts may be different for other service industries, such as hotel or retail. Therefore, the generalizability of the findings is limited. Future studies should broaden the research contexts to cross industries.

Second, this study has proposed and empirically tested the relationship of transformational leadership, psychological empowerment, and emotional labor. However, the boundary conditions of this relationship are neglected. For example, recent research has demonstrated that the relationship between transformational leadership and psychological empowerment is moderated by organizational structures [84]. Thus, future studies should explore boundary conditions that may moderate the relationship between transformational leadership, psychological empowerment, and emotional labor.

Third, given that the findings of this study suggest a partial mediation between transformational leadership and deep acting, future research should consider other alternative mediating mechanisms. For example, because transformational leaders are more likely to encourage their followers to engage job crafting to enhance their performance [85], it is reasonable to expect that job crafting might mediate the relationship between transformational leadership and deep acting. Therefore, future studies should extend the current findings and explore other potential mediators.

## 6. Conclusions

Drawing on the conservation of resources (COR) theory, the current study has proposed and tested transformational leadership’s effects on the emotional labor of front-line employees via psychological empowerment. These hypotheses were tested using data of 436 front-line employees from call centers. The results reveal that transformational leadership shows positive and negative effects on deep acting and surface acting, respectively. Psychological empowerment exerts mediating effects on the relationship between transformational leadership and front-line employees’ emotional labor. Specifically, three dimensions of psychological empowerment (impact, self-determination, self-efficacy) partially mediate transformational leadership’s effects on deep acting. Two dimensions of psychological empowerment (impact, self-efficacy) fully mediate the relationship between transformational leadership and surface acting.

## Figures and Tables

**Figure 1 ijerph-20-01030-f001:**
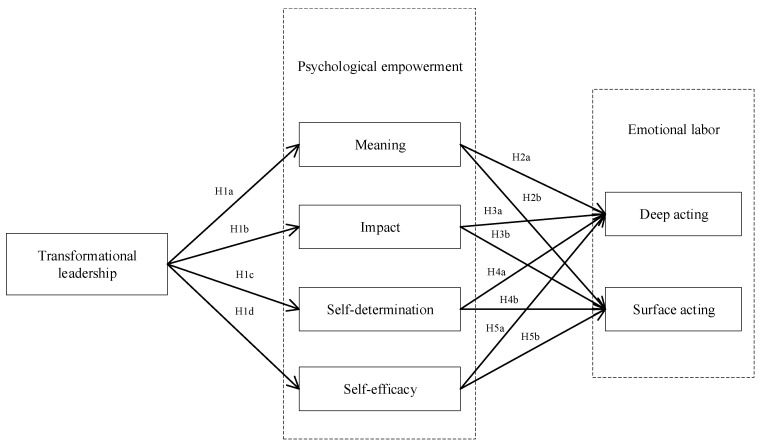
The conceptual model.

**Table 1 ijerph-20-01030-t001:** Mean, standard deviations, correlations, Cronbach alpha coefficient, and AVE.

	M	SD	ɑ	1	2	3	4	5	6	7
1.Transformational leadership	3.668	0.747	0.932	0.642						
2.Meaning	3.848	0.765	0.908	0.431 ***	0.879					
3.Impact	3.250	0.893	0.872	0.461 ***	0.273 ***	0.835				
4.Self-determination	3.658	0.853	0.862	0.490 ***	0.393 ***	0.574 ***	0.823			
5.Self-efficacy	3.663	0.986	0.894	0.425 ***	0.324 ***	0.277 ***	0.309 ***	0.862		
6.Surface acting	3.654	0.892	0.900	−0.280 ***	−0.204 ***	−0.275 ***	−0.149	−0.343 ***	0.869	
7.Deep acting	3.704	0.935	0.898	0.474 ***	0.223 ***	0.391 ***	0.456 ***	0.311 ***	−0.035	0.867

Note: The square roots of AVE are presented in diagonal elements (bold values). n = 436; *: *p* < 0.05; **: *p* < 0.01; ***: *p* < 0.001.

**Table 2 ijerph-20-01030-t002:** The results of confirmatory factor analysis.

Model	$\chi^{2}$	df	$\chi^{2}$ / df	RMSEA	CFI	TLI
7 factors: TL, Mean, IMP, SD, SE, SA, DA	1705.791	633	2.695	0.062	0.904	0.894
3 factors: TL, Mean + IMP + SD + SE, SA + DA	4499.657	651	6.912	0.117	0.656	0.629
2 factors: TL + Mean + IMP + SD + SE, SA + DA	5176.653	653	7.927	0.126	0.596	0.565
1 factor: TL + Mean + IMP + SD + SE + SA + DA	5643.985	654	8.630	0.133	0.554	0.521

Note: n = 436; TL = transformational leadership; Mean = meaning; IMP = impact; SD = self-determination; SE = self-efficacy; SA = surface acting; DA = deep acting.

**Table 3 ijerph-20-01030-t003:** The results of mediation tests.

Independent Variable		M-1	M-2	M-3	M-4	M-5	M-6	M-7	M-8
		DA	Mean	IMP	SD	SE	DA	SA	SA
Control Variables	Gender	−0.105 *	0.070	−0.053	−0.061	0.058	−0.092 *	−0.091	−0.078
	Age	−0.076	0.082	−0.001	0.014	0.082	−0.085	0.026	0.048
	Education	−0.031	−0.069	0.049	−0.033	−0.064	−0.025	0.008	0.000
	Tenure	−0.068	0.064	−0.050	−0.036	0.040	−0.058	−0.061	-0.053
TL		0.465 ***	0.431 ***	0.406 ***	0.462 ***	0.417 ***	0.304 ***	−0.250 ***	−0.094
Mean							−0.032		−0.059
IMP							0.106 *		−0.172 **
SD							0.191 ***		0.079
SE							0.111 *		−0.232 ***
R^2^		0.250 ***	0.198 ***	0.174 ***	0.213 ***	0.180 ***	0.313 ***	0.0767 ***	0.152 ***

Notes: TL = transformational leadership; Mean = Meaning; IMP = Impact; SD = self-determination; SE = self-efficacy; DA = deep acting; SA = surface acting; * *p* < 0.05; ** *p* < 0.01; *** *p* < 0.001.

**Table 4 ijerph-20-01030-t004:** Total, direct, and indirect effects of TL on deep acting.

		Effect	Boot SE	Boot LLCI	Boot ULCI
Total effects		0.582	0.053	0.477	0.686
Direct effects		0.380	0.064	0.254	0.506
Indirect effects	TOTAL	0.202	0.060	0.088	0.327
	TL-Mean-DA	−0.017	0.029	−0.075	0.038
	TL-IMP-DA	0.054	0.031	0.005	0.116
	TL-SD-DA	0.110	0.040	0.040	0.198
	TL-SE-DA	0.055	0.033	0.006	0.123

Notes: TL = transformational leadership; Mean = Meaning; IMP = Impact; SD = self-determination; SE = self-efficacy; DA = deep acting.

**Table 5 ijerph-20-01030-t005:** Total, direct, and indirect effects of TL on surface acting.

		Effect	Boot SE	Boot LLCI	Boot ULCI
Total effects		−0.298	0.056	−0.409	−0.188
Direct effects		−0.113	0.068	−0.246	0.021
Indirect effects	TOTAL	−0.186	0.061	−0.311	−0.072
	TL-Mean-SA	−0.030	0.030	−0.089	0.030
	TL-IMP-SA	−0.083	0.035	−0.158	−0.021
	TL-SD-SA	0.043	0.038	−0.029	0.124
	TL-SE-SA	−0.116	0.035	−0.190	−0.055

Notes: TL = transformational leadership; Mean = meaning; IMP = impact; SD = self-determination; SE = self-efficacy; SA = surface acting.

**Table 6 ijerph-20-01030-t006:** Hypothesis test results.

Number	Hypothesis	Result
H1a	Transformational leadership is positively related to meaning.	support
H1b	Transformational leadership is positively related to impact.	support
H1c	Transformational leadership is positively related to self-determination.	support
H1d	Transformational leadership is positively related to self-efficacy.	support
H2a	Meaning is positively related to deep acting.	reject
H2b	Meaning is negatively related to surface acting.	reject
H3a	Impact is positively related deep acting.	support
H3b	Impact is negatively related to surface acting.	support
H4a	Self-determination is positively related deep acting.	support
H4b	Self-determination is negatively related to surface acting.	reject
H5a	Self-efficacy is positively related deep acting.	support
H5b	Self-efficacy is negatively related to surface acting.	support

## Data Availability

Not applicable.

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
