# Peer review of "Transformational Leadership and Emotional Labor: The Mediation Effects of Psychological Empowerment"

_ijerph, 2023, doi:10.3390/ijerph20021030_

Round 1
Reviewer 1 Report
Dear Authors,
The article is very interesting and deals with extremely important problem both from the perspective of theory and practice. In order to increase its value, I propose to refine two areas:
1. Expanding the discussion of research results. At the moment, there is no actual discussion of your results with the results of the literature review presented in section 2. You go directly to the theoretical and practical implications.
2. Expanding the section presenting the limitations and future research as your study can suggest much more interesting future research pathways.
Author Response
Dear reviewer 1,
Thank you for giving us the opportunity to submit a revised draft of our manuscript titled “Transformational leadership and emotional labor: the mediation effects of psychological empowerment” (Manuscript ID: ijerph-2130953). We appreciate your time and efforts dedicated to providing your insightful comments on our manuscript. We have been able to incorporate changes to reflect most of the suggestions. All of our modifications are highlighted by yellow in the revised manuscript.
Here is a point-by-point response to the comments and concerns.
Point 1: Expanding the discussion of research results. At the moment, there is no actual discussion of your results with the results of the literature review presented in section 2. You go directly to the theoretical and practical implications.
Response 1: We are grateful for the suggestion. Necessary change in the discussion section 5.1 Contributions to Theory has been made in the revised manuscript. We combined the results of our study with the results of the literature review presented in section 2 according to your comment. And we illustrate the connection and distinction between the results of this study and the results of previous studies.
“Although previous studies have paid attention to the effects of transformational leadership on front-line employees’ emotional labor, such as Luo et al. [13], the psychological mechanism of these effects remain unclear. Just as Grandey et al. [62] mentioned in their work that front-line employees’ emotional labor is form of discretionary behavior, which is driven more by intrinsic motivation.” (Line 416-420, Page 10)
“This study supplements Luo and Guchait’s work [13] which merely examine the direct effect of transformational leadership on emotional labor” (Line 424-426, Page 11)
“Despite an array of prior studies have demonstrated that transformational leadership is an effective leadership style for bootstring proactive employee behaviors, such as OCB [81] and innovation behaviors [58,60], only a few studies recently turned their interests to examine transformational leadership’s effects on front-line employees’ emotional labor [13], which is a very common form of proactive behavior during their daily work. Not only does this study confirm transformational leadership’s effects on front-line employees’ emotional labor in the context of service, but this study also illustrates the psychological process of these effects by examining psychological empowerment’s mediating effects. These findings answer Siangchokyoo’s [82] call for more detailed studies to examine the role of empowerment plays during the link of transformational leadership and employees’ behavior” (Line 428-438, Page 11)
“Although prior studies have confirmed psychological empowerment, as an important psychological construct, is a typical consequence of transformational leadership [82], only a few studies have further extended this link to employees’ behavior. The results of this study confirm the existence of the logical path: transformational leadership-psychological empowerment-emotional labor” (Line 439-444, Page 11)
Point 2: Expanding the section presenting the limitations and future research as your study can suggest much more interesting future research pathways.
Response 2: We agree with the comment and re-wrote the limitations and future research section (Line 488-508, Page 12). We propose three limitations of our research: (1) the generalizability of our findings. We suggest future studies broaden the research context to cross industries.
“Although this study makes the aforementioned theoretical and practical contributions, there are also several limitations. First, the research context of this study focusses on call centers. Despite the fact that emotional labor is very common for call center employees, employees’ emotional labor strategies or efforts may be different for other service industries, such as hotel, retailing. Therefore, the generalizability of the findings is limited. Future studies should broaden the research contexts to cross industries.” (Line 489-494, Page 12).
(2) the boundary conditions of the conceptual model are unclear. We suggest future studies explore boundary conditions by introducing moderators which may influence the relationship between transformational leadership, psychological empowerment and emotional labor.
“Second, this study has proposed and empirically tested the relationship of transformational leadership, psychological empowerment and emotional labor. However, the boundary conditions of this relationship are neglected. For example, recent research has demonstrated that the relationship between transformational leadership and psychological empowerment is moderated by organizational structures [89]. Thus, future studies should explore boundary conditions that may moderate the relationship between transformational leadership, psychological empowerment and emotional labor.” (Line 495-501, Page 12).
(3) future research should consider other alternative mediating mechanisms by which transformational leadership can influence emotional labor, such as job crafting.
“Third, given that the findings of this study suggest a partial mediation between transformational leadership and deep acting, future research should consider other alternative mediating mechanisms. For example, because transformational leaders are more likely to encourage their followers to engage job crafting to enhance their performance [90], it is reasonable to expect that job crafting might mediate the relation-ship between transformational leadership and deep acting. Therefore, future studies should extend the current findings and explore other potential mediators.” (Line 502-508, Page 12).

Reviewer 2 Report
Very interesting paper. I really like the theoretical and methodological soundness of your paper. I just have two suggestions -
1. Please add a paragraph on context of the research.
2. In the conclusion section, you can add a diagrammatic summary of your findings. It can be drawn similar to the conceptual model but this one shows which of the hypothesis were proved or disproved.
Author Response
Dear reviewer 2,
Thank you for giving us the opportunity to submit a revised draft of our manuscript titled “Transformational leadership and emotional labor: the mediation effects of psychological empowerment” (Manuscript ID: ijerph-2130953). We appreciate your time and efforts dedicated to providing your insightful comments on our manuscript. We have been able to incorporate changes to reflect most of the suggestions. All of our modifications are highlighted by green in the revised manuscript.
Here is a point-by-point response to the comments and concerns.
Point 1: Please add a paragraph on context of the research.
Response 1: Thanks for your kind advice. We have added content on the research context in section 3.1 Sampling and data collection.
“In order to test the hypotheses, we carried a survey to collect data from front-line employees of call centers. The call center is selected as our study context for its character of intense employee-customer interaction, which require front-line employees to display positive emotions during the service encounter [77]. After selecting 5 call centers which located in the northwestern China, total 500 questionnaires were distributed on site to front-line employees with 100 questionnaires for each call center. The questionnaire is totally anonym and takes participants about 15 minutes to complete. When participants finish the survey, they are required to seal the questionnaire in an envelope by themselves and give it to the research assistants on site.” (Line 273-282, Page 6)
Point 2: In the conclusion section, you can add a diagrammatic summary of your findings. It can be drawn similar to the conceptual model but this one show which of the hypothesis were proved or disproved.
Response 2: We are grateful for the suggestion. We have added Table 6 (Page 10) to summarize our findings, which can provide a clear view of which hypotheses are supported or rejected.
Table 6. Hypothesis test results
|
Number |
Hypothesis |
Result |
|
H1a |
Transformational leadership is positively related to meaning. |
support |
|
H1b |
Transformational leadership is positively related to impact. |
support |
|
H1c |
Transformational leadership is positively related to self-determination. |
support |
|
H1d |
Transformational leadership is positively related to self-efficacy. |
support |
|
H2a |
Meaning is positively related to deep acting. |
reject |
|
H2b |
Meaning is negatively related to surface acting. |
reject |
|
H3a |
Impact is positively related deep acting. |
support |
|
H3b |
Impact is negatively related to surface acting. |
support |
|
H4a |
Self-determination is positively related deep acting. |
support |
|
H4b |
Self-determination is negatively related to surface acting. |
reject |
|
H5a |
Self-efficacy is positively related deep acting. |
support |
|
H5b |
Self-efficacy is negatively related to surface acting. |
support |
Thank you again for your advice. Those comments are valuable and helpful for revising and improving our paper. We look forward to hearing from you in due time regarding our submission and to responding to any further questions and comments you may have.

Reviewer 3 Report
Dear authors,
It was my pleasure to review the manuscript entitled “Transformational leadership and emotional labor: the mediation effects of psychological empowerment”. The effort invested in the preparation of the article can be clearly seen, with a good result in terms of writing and organisation of the content. Thus, I personally believe that this paper is insightful with significant information in understanding this field of study. I belief that it will be very useful to better appreciate the field of study. However, I think that the paper needs some further improvements:
· Abstract: It would be interesting to include a line on the method used (instruments) to clarify what has been done.
· Introduction: Comprehensive, with recent and relevant quotations.
· Methodology: Use of an experimental design appropriate to the stated research objectives, with clear, obvious and detailed explanations. However, I would introduce a short paragraph or even a table en el que se detallen las características de la muestra.
· Findings/Results: Comprehensive and organised. The hypotheses put forward are clearly answered.
· Discusion: It needs to be strengthened. It would be interesting to describe the most relevant results relating them to previous scientific literature on the topic and to the introduction of this study. I believe that for the study to be even more valid, this discussion between the results of this study and the results of previous studies is necessary.
· Future lines of research and implications: Fine with them.
· Implications: Finally, should have a standalone section emphasizing on implications of this study to practice and society.
· Tables and figures: They are appropriate and very clear. They are easy to interpret and they support the arguments.
Finally, congratulations once again to the authors for a very interesting article that, with some minor changes, it would be suitable for publication.
Author Response
Dear reviewer 3,
Thank you for giving us the opportunity to submit a revised draft of our manuscript titled “Transformational leadership and emotional labor: the mediation effects of psychological empowerment” (Manuscript ID: ijerph-2130953). We appreciate your time and efforts dedicated to providing your insightful comments on our manuscript. We have been able to incorporate changes to reflect most of the suggestions. All of our modifications are highlighted by blue in the revised manuscript.
Here is a point-by-point response to the comments and concerns.
Point 1: Abstract: It would be interesting to include a line on the method used (instruments) to clarify what has been done.
Response 1: We agree with the comment and re-wrote the sentence in the revised manuscript as the following:
“By collecting data from 436 employees in five call centers, we test our model and hypotheses through PROCESS 3.3 macro for SPSS developed by Hayes, the results show that: transformational leadership shows positive and negative effects on deep acting and surface acting, respectively.” (Line 13-16, Page 1)
Point 2: Methodology: Use of an experimental design appropriate to the stated research objectives, with clear, obvious and detailed explanations. However, I would introduce a short paragraph or even a table en el que se detallen las características de la muestra.
Response 2: Thanks for your kind advice. We have added a paragraph in the section 3.1Sample and procedures to describe the samples we collected. Specifically,
“the sample consisted of 436 employees (man=30, woman=406). The number of employees between 25 and 30 years is the largest (23.6%). And 78% of employees had received junior college education or above. In terms of the tenure, the largest number of groups is one to three years (36.7%).” (Line 284-287, Page 7)
Point 3: Discussion: It needs to be strengthened. It would be interesting to describe the most relevant results relating them to previous scientific literature on the topic and to the introduction of this study. I believe that for the study to be even more valid, this discussion between the results of this study and the results of previous studies is necessary.
Response 3: We are grateful for the suggestion. In the revised manuscript, we extend the discussion section. First, we added paragraph to summarize the purpose and findings of this study.
“The current study proposed and tested a conceptual model to explore the mechanism by which transformational leadership influence emotional labor strategies of front-line employees. Our results demonstrate that transformational leadership exerts positive and negative effects on deep acting and surface acting, respectively. Psychological empowerment serves as a mediator of the linkage of between transformational leadership and two emotional labor strategies. Specifically, the positive effect of transformational leadership on deep acting is partially mediated by three dimensions of psychological empowerment: impact, self-determination and self-efficacy. While the negative effect of transformational leadership on surface acting is fully mediated by two dimensions of psychological empowerment: impact and self-efficacy.” (Line 403-412, Page 10)
Second, we introduced some relevant results of previous literature on the topic and contrasted them with our findings, thereby enhancing the value of our research. Meanwhile, we further expand our findings by extending the contribution to the literature. By doing so, the discussion section could become more reasonable and convincing.
“Although previous studies have paid attention to the effects of transformational leadership on front-line employees’ emotional labor, such as Luo et al. [13], the psychological mechanism of these effects remain unclear. Just as Grandey et al. [62] mentioned in their work that front-line employees’ emotional labor is form of discretionary behavior, which is driven more by intrinsic motivation.” (Line 416-420, Page 10)
“This study supplements Luo and Guchait’s work [13] which merely examine the direct effect of transformational leadership on emotional labor” (Line 424-426, Page 11)
“Despite an array of prior studies have demonstrated that transformational leadership is an effective leadership style for bootstring proactive employee behaviors, such as OCB [81] and innovation behaviors [58,60], only a few studies recently turned their interests to examine transformational leadership’s effects on front-line employees’ emotional labor [13], which is a very common form of proactive behavior during their daily work. Not only does this study confirm transformational leadership’s effects on front-line employees’ emotional labor in the context of service, but this study also illustrates the psychological process of these effects by examining psychological empowerment’s mediating effects. These findings answer Siangchokyoo’s [82] call for more detailed studies to examine the role of empowerment plays during the link of transformational leadership and employees’ behavior” (Line 428-438, Page 11)
“Although prior studies have confirmed psychological empowerment, as an important psychological construct, is a typical consequence of transformational leadership [82], only a few studies have further extended this link to employees’ behavior. The results of this study confirm the existence of the logical path: transformational leadership-psychological empowerment-emotional labor” (Line 439-444, Page 11)
Point 4: Implications: Finally, should have a standalone section emphasizing on implications of this study to practice and society.
Response 4: Thanks for your kind advice. After revising, we expand the implications of our study to practice and society and make some recommendations to service firms, mangers and front-line employees, respectively.
“Firstly, service firms have to realize the importance of leader in influencing front-line employees’ emotional labor. According to the findings of this study, transformational leadership is an ideal leadership style for front-line employees. Therefore, during the leader recruiting or promotion process, it is necessary to set criteria considering candidates ‘competence or personality which make them easier to perform transformational leadership behaviors. By doing so, service firms can guarantee the leaders are suitable for their job and can effectively exert influence on front-line employees.” (Line 457-463, Page 11).
“Secondly, service managers should alter their leadership style to transformational leadership, which has been suggested to promote front-line employees’ deep acting and reduce surface acting. Therefore, training programs for service managers should focus on the skills of engaging transformational leadership behaviors. According to the definition of transformational leadership, typical transformational leadership behaviors include idealized influence, intellectual stimulation, inspirational motivation, and individualized consideration [84]. Once leaders take these behaviors in their routine work, the front-line employees tend to take deep acting rather than surface acting during service encounters. Furthermore, organizational culture also can foster climate suitable for transformational leadership. If the organizational culture encourages employee self-growth, providing employee future vision, caring employee’s needs and well-binges, the leaders are prone to take transformational leadership to comply with the organizational culture.” (Line 464-475, Page 11)
“Thirdly, efforts should be devoted to enhance front-line employees’ psychological empowerment. The results of this study show that psychological empowerment plays an important role in the link of transformational leadership and emotional labor. In order to enhance front-line employees’ psychological empowerment, service firms should especially improve employees’ sense of impact. For example, service firms can send signals of caring and valuating employees’ contribution by providing positive feedback, such as rewarding. To improve front-line employees’ self-determination, managers can invite front-line employees to take part in the decision making. Also, enhancing front-line employees’ job autonomy through empowering is an effective way to improve front-line employees’ self-determination. In addition, enhancing self-efficacy is necessary to eliminate front-line employees’ surface acting. Managers should encourage employees to set challenging goals and provide supports to help them to overcome difficulties and complete the tasks”. (Line 476-487, Page 12)
Thank you again for your advice. Those comments are valuable and helpful for revising and improving our paper. We look forward to hearing from you in due time regarding our submission and to responding to any further questions and comments you may have.
